# Psychosocial safety climate (PSC) and working conditions, predictors of mental health and antidepressant and opioid use in Australia: a study protocol for longitudinal data linkage

Cherie Natalie Crispin ![ORCID],[1] Ali Afsharian,[1] May Young Loh,[1] Maureen F Dollard,[1] Christian Dormann,[2] Nick Glozier ![ORCID],[3] Tiffany Gill ![ORCID],[4] Anne W Taylor[5]

¹Justice & Society, University of South Australia, Adelaide, South Australia, Australia
²Johannes Gutenberg Universitat Mainz, Mainz, Rheinland-Pfalz, Germany
³University of Sydney Brain and Mind Research Institute, Sydney, New South Wales, Australia
⁴Medical Specialities, The University of Adelaide, Adelaide, South Australia, Australia
⁵School of Medicine, The University of Adelaide, Adelaide, South Australia, Australia

**Correspondence to**
Cherie Natalie Crispin;
cherie.crispin@unisa.edu.au

## ABSTRACT

**Introduction** Work-related stress is a social determinant of global health that represents a huge cost to workers' health and reduces work performance. In Australia, mental well-being is a pressing national issue—with one in five Australians experiencing mental disorders. Antidepressants are a first-line medication commonly used to treat mental disorders. Recently, Australia has seen a dramatic increase in the use of prescribed antidepressant medications to treat mental health related illnesses. Australia has also seen a dramatic increase in the use of prescribed opioid analgesics for non-cancer pain including opioid use for psychological distress and social stressors. It is plausible a rise in mental health problems and antidepressant and opioid medication use is partly attributable to the corporate climate for worker mental health (ie, the psychosocial safety climate, PSC). This research aims to identify how PSC and workplace conditions contribute to employee well-being and distress that culminate in antidepressant and opioid medication use.

**Methods/analysis** Data will be collected through creative data linkage from the Australian Workplace Barometer (AWB), to medication data (via the Pharmaceutical Benefits Scheme, PBS). The participant sample will include 1372 working Australians from the AWB project from 2009 to 2021. Four waves of longitudinal data from 2009 to 2021 will be used to investigate the plausible link between Australia's high levels of antidepressant and opioid use and distress at work. The project advances theory by probing the role corporate climate plays in work design, distress, mental health problems and antidepressant and opioid use. It will determine if antidepressant and opioid use has led to an underestimation of work stress effects. Proposed theoretical models will be analysed through linked data, using continuous time structural equation modelling, hierarchical linear modelling, logistic regression and cost estimation.

**Ethics and dissemination** The study has been approved by the Human Research Ethics Committee of the University of South Australia (Ethics Protocol: 203003). Further, approval from the Australian Institute of Health and Welfare Ethics Committee was also granted for linkage of AWB data and PBS data (EthOS Application EO2022/1/1190).

### STRENGTHS AND LIMITATIONS OF THIS STUDY

⇒ This study is the first large-scale longitudinal data linkage to investigate the role of PSC in linking work stressors, mental health, and medication use in the Australian working population.
⇒ The longitudinal linkage of both data sources enables information to be chronologically ordered to provide valuable insight otherwise missing from single source cross-sectional data.
⇒ Data linkage will overcome common method bias associated with self-report research of mental health, through validation with objective measures.
⇒ Data linkage success is largely dependent on specific individual participant characteristics (eg, first name, surname, date of birth) provided at the point of interview. As this information was limited, this restricted a more robust data linkage.
⇒ Due to the nature of data privacy, the data linkage process can be prolonged due to; the development of technical documentation pertaining to specific use of the data; the provision of a secure workspace with encrypted data and strict personnel access; as well as the rigorous process of ensuring all identifiable data are removed from both data sources.

Results of the study will be disseminated through worldwide keynotes, key international settings, high-impact peer-reviewed journals, industry conference presentations and media outlets to reach managers, workers, and industry partners. Further, UniSA requires publications from public projects to be held in an institutional repository which fulfils the Australian Research Council's Open Access Policy.

## BACKGROUND

Work-related stress is a social determinant of global health[1] that represents a huge cost to workers' health and reduces work performance.[2] Every year an estimated 12 billion working days are lost to depression and anxiety, with approximately 15% of

working-age adults worldwide reported to have a mental disorder in 2019.[3] In Australia alone, one in five Australians are experiencing mental disorders[4] with problems related to employment the most common risk factors of suicide in the working age population.[5]

A common form of treatment for mental disorders is antidepressant medication.[6] Epidemiological studies have demonstrated an increase in usage of antidepressant prescriptions in industrialised countries during the 21st century.[6 7] Since 2017, Australia has seen a dramatic increase in the use of prescribed antidepressant medications to treat mental health-related illnesses.[8] The Australian Institute of Health and Welfare (AIHW) reported that in 2020–2021, 73.1% of mental health-related prescriptions were filled for antidepressant medications.[8] Additionally, over the past two decades, Australia has seen a dramatic increase in the use of prescribed opioid analgesics for non-cancer pain including opioid use for psychological distress and social stressors.[9 10] According to the AIHW,[11] in 2019 opioid use within Australia (3.3% of the adult population) remained higher than the global average (1.2% of the global population aged 15–64 years). It is plausible a rise in mental health problems and antidepressant and opioid medication is partly attributable to the corporate climate. Indeed, only 52% of employees across Australia think their workplace positively supports their mental well-being,[12] and only 9% of employers have an 'integrated and sustained approach to mental health and wellbeing'.[13] Work pressure and bullying—products of poor management—account for almost 7500 workers compensation claims per annum for work-related mental disorders.[14]

During the past few decades, a multitude of studies have been published on working conditions and mental health and prescribed medication use.[15–17] Yet the theory and evidence linking work to mental ill-health and medication use are incomplete in significant ways. First, studies largely focus on work design, such as the levels of job demands (eg, work pressure, emotional demands such as bullying) and resources (eg, job control, social support) and their combinations as precursors to worker ill health and increased risk of depressive symptoms.[16] Although there is a strong theoretical (Job Demand-Control-Support (DCS)[18]; and the Effort-Reward Imbalance (ERI) model[19]) and empirical base linking work design to mental health problems,[16 20 21] prominent researchers are now going further and asking, "Where do work design and work conditions come from?".[22] Dollard *et al*[23–25] believe that the root cause emanates further upstream, that is with the corporate climate (i.e. psychosocial safety climate, PSC).

PSC theory is an innovation in the field,[23–25] defined as 'policies, practices and procedures for the protection of worker psychological health and safety' (23, p579). It refers to management commitment and priority, organisational communication and organisational participation and involvement, specifically in relation to worker psychological health and safety. PSC theory has gained prominence nationally and internationally and is a unifying construct that has promoted interdisciplinarity integrating work stress and safety science research. The construct is empirically distinct from related constructs such as team psychological safety, perceived organisational support and safety climate.[26] Whereas the safety climate construct is used to predict safety behaviour and accidents and injuries,[27] PSC instead is used to predict psychosocial work design and psychological health and is also linked to the enactment of safety behaviours and the reporting of workplace injuries.[28]

Work design theories are nevertheless important to explain the process and why PSC relates to mental health outcomes. A comprehensive work design framework, such as the Job Demands-Resources model,[29] is required that integrates the components of ERI and DCS models, by encompassing various kinds of demands and resources in explanation of mental health and engagement. Work design is influenced by the organisational priorities and policies. Tying PSC to work design theories (eg, the JD-R model), whereby PSC is the predictor of the JDR model is likely a feasible explanatory framework to understand the relationship between work conditions associated with work stress reactions and mental health problems linked to antidepressant and opioid medication use. The PSC extended JDR theory is the backdrop for this study.

This research will yield evidence to stimulate corporate climate change to protect workers' psychological health and well-being. The Australian Workplace Barometer (AWB) has connected work factors to self-reported mental ill-health.[30] However, it has not yet been investigated whether these contribute to the high levels and costs of antidepressant and opioid use. Moreover, it is unknown whether the large consumption of antidepressants and opioids by Australians is masking a more urgent work stress problem by reducing symptoms leading to underestimation in studies. Uncovering the 'true mental effects' of stressful work is essential.

## Aims

This research aims to identify how the corporate climate (i.e. PSC) and workplace conditions contribute to Australian employee well-being and distress that culminate in them using antidepressant and opioid medications. This study will investigate the plausible link between distress at work and Australia's high levels of antidepressant and opioid use through the creative linkage of data from the AWB (10-year longitudinal study) to antidepressant and opioid medication data (via the data collected by Australian national Pharmaceutical Benefits Scheme (PBS; which contains information on prescribed medicines that qualify for medical claims). The project advances theory by probing corporate climate's role in work design, distress, mental health problems, stigma/discrimination and antidepressant and opioid use. It will determine if antidepressant and opioid use has led to underestimating work stress effects. Data will also be used to estimate the $AUD cost of work-related antidepressant and opioid use.

## Research questions
### Research question 1
The main research question is 'Does Psychosocial Safety Climate have a primary role, in process pathways linking working conditions to mental health problems and anti-depressant/opioid medication use?'

### Research question 2
What is the cost (in $AUD) of antidepressant/opioid medication use related to work factors?

### Research question 3
Is there a positive relationship between Australian employees' experience of stressful work conditions (eg, high demands, low resources, bullying, harassment, stigma, discrimination) and antidepressant and opioid medication use.

### Research question 4
Is there a positive relationship between Australian employees' experience of distress, emotional exhaustion and depressive symptoms and antidepressant and opioid medication use?

### Research question 5
Does antidepressant/opioid use lead to an underestimation of the impact of stressful work on employees?

### This research will include two separate studies:
Study 1: will look at corporate climate, work stressors, mental health and *antidepressant* medications.

Study 2: will look at corporate climate, work stressors, mental health and *opioid* medications.

## METHODS AND ANALYSIS
### Study design
This study will use a longitudinal design that consists of four waves of interview and self-report data using an existing data set, the AWB, a population-based longitudinal data set, matching individuals over time and linking to objective medication data (PBS). The four waves of data from the AWB dataset include data collected in 2009–2010 (wave 1), 2010–2011 (wave 2), 2014–2015 (wave 3) and 2020–2021 (wave 4). The AWB cohort data will be linked to PBS data collected from 1 January 2009 to the latest available data. The different time points between data collection will help track the progression of effects across different time periods. The current study will commence in May 2023 and run until December 2023.

### Explanation of existing data
As of the submission date, the data exists, and researchers have only had access to the AWB data. However, no detailed analyses have been conducted concerning the current research plan (including the calculation of summary statistics). Access to AWB data have been limited, though a small portion of the data has been used to conduct exploratory analysis that showed encouraging initial results. PBS data have not been accessed at the point of registration.

### Data collection procedures
In the first round of data collection, the AWB used a population-based random sampling approach to conduct computer-assisted telephone interviews. Participants were recruited randomly from the Australian electronic white pages inviting household members to participate (the participant had to have had the last birthday in the household to be eligible).

For each subsequent wave of data collection, those who had consented to be recontacted were sent a letter advising of further data collection and recontact for a further interview. A combination of modes was used to collect subsequent waves of AWB data to increase sample representativeness and participation uptake, including telephone interviews by landline or mobile, surveys via email or SMS link, and hard copy surveys sent via post. Where participants could not partake in interviews at the first point of contact, call-backs for a more suitable time were arranged.

Telephone interviews were conducted by a team of research assistants trained to enter data immediately into the Qualtrics online survey. Participants who received an email or SMS link were provided with their participant ID (required for linkage) and entered their participant ID and self-reported data directly into the Qualtrics survey link provided. Hard copy surveys were entered manually into the Qualtrics online survey by the research team on receipt.

Detailed information on survey items used in the study are uploaded as supplementary information (see online supplemental file S1).

### *Payment for participation*
In the fourth wave of data collection, participants who completed the survey automatically went into the draw to receive a gift card valued at $A100. The winner of the draw was picked using a random number generator, contacted by the research team's chief investigator by phone, and announced the winner.

### *Inclusion criteria*
Participants are required to be a minimum of 18 years of age and currently in paid employment. Subcontractors must have worked with the same employer for a minimum of 4 months. Participants must have consented to their personal AWB information being securely sent to the AIHW to match the PBS dataset for research purposes.

### *Exclusion criteria*
Ages out of the age range, unemployed and those who did not consent to data linkage are excluded from the dataset.

### *PBS dataset*
National Medicare and PBS will provide data on prescription medicines, defined by the AIHW project

as antidepressants to treat clinical depression and opioid medication use for this project. The PBS data comprises individual antidepressant and opioid medication use (summed over a year) from 2009 to 2021.

### Data linkage (AWB linkage to PBS)

PBS data were linked via AWB participants who consented to have their AWB data linked to PBS data in 2020/2021. Antidepressant and opioid medication will be matched with the AWB at the individual level, using individual names and addresses.

### Data linkage procedure

Step 1: A 'Request for linkage of national data: technical assessment' was completed by the second author as part of the AIHW data linkage protocol and AIHW ethics approval, which outlined the proposed research and study aims.

Step 2: AIHW was sent two files via secure email, including:
► AWB cohort personal identifiers with a data ID code.
► AWB content data and with the data ID code.

Step 3: AIHW linked the AWB cohort to PBS. This linkage process included the following:
► Creation of a new project-specific person number.
► Loading the PBS content data and the AWB content data to the Secure Unified Research Environment (SURE) project-specific person number.
► All data transfers were via secure messenger service.

Important data linkage notes:
► The technical assessment is a rigorous process required by AIHW to ensure that the data requested for the linkage answer the proposed research questions. The technical assessment ensures the study aims are achievable using the proposed linkage methodology and forms part of the AIHW ethics approval for the research project.
► AIHW set up a specific project ID for the linkage of the AWB and PBS data and the provision of a secure workspace, referred to as the SURE workspace, for the analysis of linked data. All analytical software packages (SPSS, R) are loaded onto the SURE secure workspace. Deidentified AWB data set and PBS data are curated into the secure workspace.
► There are strict protocols in place during SURE access, where the original AWB dataset is held on a secure university server that restricts researcher access involved with SURE data analysis. This highly confidential workspace is constructed to hold data that blocks all access to other desktop files and software to avoid identifying deidentified participants.
► A research assistant (the first author) will be the data custodian (not involved in the data analysis) and will be appointed during this time to ensure no local data access of identifiable/reidentifiable data, while there is access to SURE data. All data analysis takes place with the data on the secure SURE website.

### Sample size

The sample included 1372 working Australians who participated in the AWB study from 2009 to 2021 and consented to the data linkage. Participants are located across all Australian states and territories; South Australia (SA), New South Wales (NSW), Western Australia (WA), Victoria (VIC), Queensland (QLD), Tasmania (TAS), Australian Capital Territory (ACT), and the Northern Territory (NT).

### Patient and public involvement

There was no patient or public involvement in the design and development of the study.

### Measured variables

All independent and dependent variables are measured at the individual level.

### Demographics and workplace information

Demographic variables in the current study include age, gender, industry, education and income. Work-related information includes work hours (ie, shift work), length of service, working status (employed vs unemployed) and employment status (permanent full-time, permanent part-time, casual and fixed term contract), occupation, size of organisation, organisation business type (ie, state government, not-for-profit) name of current organisation (optional), and lodgement of workers compensation claims.

### Psychosocial safety climate

The PSC was measured using a scale consisting of 12 items (PSC-12)[31] that assess four dimensions of PSC, including management priority, management commitment, organisational communication and organisational participation. Items were measured on a 5-point Likert scale, ranging from 1 (strongly disagree) to 5 (strongly agree). Scores for all 12 items are summed to provide an overall score for PSC. An example item is, 'In my workplace, senior management acts quickly to correct problems/issues that affect employees' psychological health'.

PSC was measured longitudinally as an independent variable.

### Psychosocial demands
#### Work pressure

Work pressure was measured using the 5-item psychological job demands scale from the Job Content Questionnaire (JCQ 2.0),[32] an example item 'My job requires working very fast'. Items were measured on a 4-point Likert scale, ranging from 1 (strongly disagree) to 4 (strongly agree). The five items were summed to give a total score for work pressure.

#### Workplace bullying

Bullying was assessed using an amended version of the QPSNordic Bullying Questionnaire.[33] Participants were provided with a definition of workplace bullying and then

answered questions about their exposure to bullying in the workplace.

Definition provided to participants; 'Bullying is a problem at some workplaces and for some workers. To label something, as bullying, the offensive behaviour has to occur repeatedly over a period of time, and the person confronted has to experience difficulties defending him or herself. The behaviour is not bullying if two parties of approximate equal 'strength' are in conflict or the incident is an isolated event'.[34]

Participants were then asked, 'Have you been subjected to bullying at the workplace during the last six months?' Those who replied with 'yes', were then asked by whom (supervisor/coworker) and the length and frequency of exposure to workplace bullying.

### Workplace harassment

Seven items from Richman *et al*'s (1996) scale[35] were used to measure workplace harassment. Items were measured on a 5-point Likert scale, ranging from 1 (very rarely/never) to 5 (very/often always) and summed together to represent a total for workplace harassment. An example item is 'I have been sworn at and/or yelled at'.

Psychosocial demands (work pressure, workplace bullying and workplace harassment) were all measured longitudinally as independent variables.

### *Psychosocial resources*
#### Job control

Three subscales were used from the JCQ 2.0[32] to measure job control, including the 5-item skill discretion subscale ('My job requires that I learn new things'), the 3-item decision authority subscale ('In my job I have very little freedom to decide how I do my work') and the 3-item macrodecision latitude subscale ('In my organization, I have significant influence over decisions made by my work team and department'). All 12 items were measured on a 4-point Likert scale, ranging from 1 (strongly disagree) to 4 (strongly agree). Items were summed to provide a total for job control.

#### Social support

To measure social support, six items from the JCQ 2.0[32] were used, three items for supervisor social support ('My supervisor/manager is helpful in getting the job done') and three items for coworker support ('People I work with are friendly'). Items were measured on a 4-point Likert scale, ranging from 1 (strongly disagree) to 4 (strongly agree) and summed for both supervisor and coworker to provide a total score for social support.

Job control (skill discretion, decision authority and macrodecision latitude) and social support (supervisor and coworker) were measured longitudinally as independent variables.

### *Distress symptoms*
#### Psychological distress

The Kessler 10 scale[36] was used to measure psychological distress. A 5-point Likert scale measured responses, ranging from 1 (none of the time) to 5 (all of the time). Scores on the 10 items were totalled to give an overall score on psychological distress. An example item is '[In the past four weeks], about how often did you feel tired out for no good reason?'.

#### Emotional exhaustion

Five items from the Maslach Burnout Inventory[37] were used to measure emotional exhaustion on a 7-point Likert scale, ranging from 1 (never) to 7 (every day). An example item is, 'I feel burned out from my work'.

Distress symptoms and/or mental health problems were measured longitudinally using psychological distress and emotional exhaustion as dependent variables.

### *Work engagement*

Work engagement was measured using three items from the Utrecht Work Engagement Scale—Shortened Version 9,[38] covering vigour, dedication and absorption. Items were measured on a 7-point Likert scale, ranging from 1 (never) to 7 (every day) and summed together to provide an overall score for work engagement. An example item is 'At work I feel bursting with energy'.

Engagement was measured longitudinally as a dependent variable.

### *Stigma/discrimination*

Stigma/discrimination was measured using one item from the Kessler 6 (K6) mental health symptom screening questionnaire ('In the last 12 months, do you think you have had emotional or mental health problems?'). If participants answered 'yes', a further seven questions were asked about their experiences within the workplace of discrimination ('Have any of your supervisors and or managers treated you unfairly or discriminated against you because of the mental health problems?'), avoidance ('Have any of your supervisors and or managers avoided you because of the emotional or mental health problems?') and positive treatment ('Because of your mental health or emotional problems have you been treated more positively by your employer?').[39]

Stigma/discrimination was only measured in wave 4 (2020/2021).

### *Mental health problems*
#### Depression

All nine items from the Patient Health Questionnaire (PHQ-9)[40] were used to measure depressive symptoms in the present study, with a time reference modified for the last 4 weeks, an example item '[During the last month, how often were you bothered by] feeling down, depressed, or hopeless?'. Items were measured on a 4-point Likert scale, from 1 (not all) to 4 (nearly every day). Scores were totalled to give an overall score on depression.

#### Suicidal ideation

One item from the PHQ-9[40] was used to measure suicidal ideation ('[During the last month, how often were you

bothered by] thoughts that you would be better off dead or thoughts of hurting yourself in some way?').

Depression and suicidal ideation are dependent variables, measured longitudinally.

### Prescribed medication usage
#### Self-report prescribed medication usage data
Participants were asked the question, 'Have you taken any prescription medications for your mental health in the last 12 months?'. Those who answered 'yes' were then asked to indicate from a list of prescribed medications (sleeping tablets/capsules, tablets/capsules for anxiety or nerves, tranquillizers, antidepressants, mood stabilisers and other medications for your mental health) which medications they had used in the last 12 months, and also in the last 4 weeks. Antidepressant medication is a dependent variable measured only in wave 4 (2020/2021).

#### Registered PBS medication data
PBS variables include PBS item (i.e. anti-depressants, opioids), drug type, benefit amount, the patient contribution amount, under copayment prescription type, prescribed specialty, prescriber type, date of prescribing, number of scripts dispensed, quantity supplied, date of supply and patient category.

### Supplemental material
S1. AWB Survey for Wave 4 (2020/2021).

### Data analysis plan
The proposed theoretical models will be analysed using data linked from the AWB and PBS, mainly using analysis of variance (ANOVA), structural equation modelling (SEM), hierarchical linear modelling (HLM), where applicable, and logistic regression.

#### Analysis of variance
By using the depression score over time, we will categorise the sample into four categories, that is, depressed, recovering, worsening and healthy to capture the changes in their mental health condition over 10 years. We will conduct ANOVA to identify the differences in their reported corporate climate and working conditions.

SEM will be used with time-lagged effects to estimate the direction and strength of the hypothesised relationships between work factors, stress reactions and medication use. We will evaluate model fit using indices like $\chi^2$ tests, root mean square error of approximation (RMSEA), comparative fit index (CFI) and Tucker-Lewis index (TLI). With time-lagged SEM, we will be able to depict how variables and their associated relationships with other variables change over time (eg, how a change in PSC would influence the change in distress reactions in the future). To better handle the varied time intervals in the AWB data, we will use continuous time SEM (CTSEM). CTSEM, with a Bayesian approach, allows individual-varying specifications for time intervals and other parameters in the analysis, addressing potential issues due to the unequal intervals between measurements and providing inferences

on continuous processes over time. We will also estimate within-person and between-person effects simultaneously by modelling both observed covariates and unobserved heterogeneity to gain a more nuanced understanding of the tranquilising effects of medication on work stress and the influence of PSC in the process pathway.

HLM will be used to account for hierarchical data structures, such as individuals nested within organisations (if the data allow) or multiple occasions within individuals. Using HLM, we will investigate how work factors and stress reactions vary at both the individual and organisation levels, and their impact on medication use.

Logistic regression will be conducted for any potential binary classification problems, such as the use of medication predicting whether workers will use medication (yes/no) based on the levels of work stress or poor organisational climate they are exposed too. ORs will be calculated to estimate the risk level of exposure to poor work conditions and low levels of PSC.

### Cost estimation
The cost associated with mental health issues and medication use attributable to work-related factors will be calculated by the total cost of medication usage and medical compensation claims through PBS data and the attributable fractions of poor work condition, PSC and work stress. AFs can be estimated, for example, by using an empirical established threshold, the PSC benchmark[41] to identify the proportion of exposed and unexposed workers to poor PSC and work condition.

### Exploratory analysis plan
We will investigate the role of stigma/discrimination and PSC for workers experiencing work stress/distress symptoms and their association with antidepressant and opioid medication use to determine whether the corporate climate tranquillizes the effects of work stress and antidepressant and opioid use and whether stigma moderates these effects. We will explore mental health vulnerability as a predictor by assessing a range of indicative states suggested by mental health self-reports. Instead of controlling for this as prior research,[42] we will investigate its trajectory, subsequent stressor exposure (eg, bullying) and future mental health status. This has implications for how mental health is managed in the workplace. We will also investigate the validity of self-report medication use data against PBS data (registered data).

## ETHICS AND DISSEMINATION
The study has been reviewed and approved by the Human Research Ethics Committee of the University of South Australia (ethics protocol: 203003). Further, approval from the AIHW Ethics Committee was also granted for the linkage of AWB data and PBS data (EthOS Application EO2022/1/1190).

Results of the study will be disseminated through worldwide keynotes, key international settings such as the Joint

Congress of International Commission on Occupational Health - Work Organisation and Psychosocial Factors (ICOH-WOPS), high-impact peer-reviewed journals, industry conference presentations and various media outlets (such as LinkedIn, Twitter, radio, community forums) to reach managers, workers and industry partners. Further, UniSA requires publications from public projects to be held in an institutional repository which fulfils the Australian Research Council's Open Access Policy. Therefore, we will also lodge relevant outputs with the Analysis and Policy Observatory collection for greater accessibility.

**Contributors** CNC was involved in collection of longitudinal data since 2020. She will assist with preparation and drafting of study papers, ethics applications, development of study protocol, development of model design and theoretical framework for Study Two (corporate climate, work stressors, mental health and opioid medications), provision of AWB data sets for data linkage agencies, liaison with data linkage agencies. AA was involved in collection of longitudinal data since 2020. He will assist with ethics applications, progress reports, drafting papers and study protocol, development of methodology, model design and theoretical framework for Study One (corporate climate, work stressors, mental health, and antidepressant medications), liaise with data linkage agencies and will be one of three researcher team members to perform computations and statistical analysis. MYL will assist with theory development, methodology and model design for Studies One and Two and will be one of three research team members to perform computations and complex statistical analysis. MFD conceived of the idea, developed and contributed to the original design and implementation of the research. MFD will lead, supervise and coordinate the research team and take overall responsibility for the strategic intellectual direction of the project and communication of results. Further, she will assist with theory development, methodology and model design for Studies One and Two and will be one of three research team members to perform computations and statistical analysis. CD will assist the team in integrating and testing new societal cost elements of Psychosocial Safety Climate (PSC) theory using R, structural equational modelling, and continuous time modelling, and contribute to publication of results. NG will be responsible for aspects of data linkage, mental health and health service evaluation, and PBS medication interpretation and will contribute to the costing and epidemiological aspects of the study design. TG is an epidemiologist who was involved in the original development and implementation of the AWB surveys in 2009. She will provide advice on survey, ethical and other project related issues as required. AWT will assist with expert advice on interpretation of data analysis and data-linkage. AWT was originally involved with sampling, the set-up of Computer Assisted Telephone Interviewing and data collection. She will assist with the review of papers where required. All authors have reviewed and approved the final version of the study protocol.

**Funding** This work was supported by five grants: (1) Australian Research Council (ARC) DP200102752 (103319) Dollard M.F. (2) DP087900 Australian Research Council Discovery Grant. Dollard MF, Winefield AH, LaMontagne AD, Taylor AW, Bakker AB, Mustard C. Working wounded or engaged? Australian work conditions and consequences through the lens of the job demands resources model. (3) LP100100449 Australian Research Council Linkage Grant. Dollard MF, Winefield AH, Taylor AW, Smith PM, Nafalski A, Bakker A, Dormann C. State, organisational, and team interventions to build a psychosocial safety climate using the Australian Workplace Barometer and the StressCafé. (4) FL200100025 Australian Research Council Laureate Fellowship. (5) Safework NSW's Centre for WHS.

**Competing interests** None declared.

**Patient and public involvement** Patients and/or the public were not involved in the design, or conduct, or reporting, or dissemination plans of this research.

**Patient consent for publication** Not applicable.

**Provenance and peer review** Not commissioned; externally peer reviewed.

**ORCID iDs**
Cherie Natalie Crispin http://orcid.org/0000-0001-6928-8442
Nick Glozier http://orcid.org/0000-0002-0476-9146
Tiffany Gill http://orcid.org/0000-0002-2822-2436

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
