## [Reviewer comments · BMJ Open]

ARTICLE DETAILS

TITLE (PROVISIONAL)	Psychosocial safety climate (PSC) and working conditions, predictors of mental health and antidepressant and opioid use in Australia: A study protocol for longitudinal data linkage.
AUTHORS	Crispin, Cherie; Afsharian, Ali; Loh, May; Dollard, Maureen F.; Dormann, Christian; Glozier, Nick; Gill, Tiffany; Taylor, Anne

VERSION 1 – REVIEW

REVIEWER	Gerding, Thomas University of Cincinnati
REVIEW RETURNED	25-Jul-2023

GENERAL COMMENTS	This sounds like a very interesting study topic. I am curious whether you can find a relationship between occupational stress exposures and antidepressant / opioid medication use. The reviewed protocol outlines your goals and objectives explicitly and to a degree appropriate for publication.
--

REVIEWER	Chudzicka-Czupała, Agata SWPS University of Social Sciences and Humanities
REVIEW RETURNED	26-Jul-2023

GENERAL COMMENTS	Thank you very much for the opportunity to learn about this inspiring, very interesting research project, which was possible through this review. Like the Authors, I am convinced that this study protocol may and should be published in the journal BMJOpen. I have essentially no substantive or ethical comments on this ambitious research project. I believe that the results of this study can lead to very important conclusions. Both its objectives, methods, sample and selection of subjects, and the potential benefits of the results are described very clearly here. The literature cited is current and relevant. The Authors have also addressed the ethical issues of the research. I wish them success in their research.
--

REVIEWER	Navinés , Ricard IDIBAPS
REVIEW RETURNED	01-Aug-2023

GENERAL COMMENTS	This is an ambitious study, which using national databases (Australian Workplace Barometer, AWB, Pharmaceutical Benefits Scheme, PBS), wants to answer through a creative method, the relationship between occupational factors, mental health and the use of antidepressants and opioids
---

	Introduction: In my opinion, the introduction is a bit disorganized. At the end of the second paragraph, an objective of the study is already introduced. It is preferable that all the objectives are discussed together at the end of the introduction, because otherwise it cuts the discourse a bit. It begins with the increase in use antidepressants, opioids, corporate climate, then depression, again theories of stress at work, corporate climate... In my opinion this part of the introduction has to be rewritten to make it better understood. Research questions: In my opinion, the main research question is not fully defined. In fact, the first research question is four. In my opinion, the main research question should be more precisely defined, and perhaps delimited a little more objectives in general, so that it is not confusing or excessive. Confirmatory analysis Plan: The statistical method to be carried out is not clearly explained Minor suggestions: Apart from the contribution of each author, it is not necessary the authors be named within the protocol. Check the tenses, and write it in the future if the work has not yet been done.
--	---

REVIEWER	Ding, Cody S. University of Missouri
REVIEW RETURNED	13-Sep-2023

GENERAL COMMENTS	Overall, this is a well-written manuscript of pre-registered research. The research questions were clear, and the arguments for the study were well articulated. The description of data sources was clear. I do not have major concerns about this pre-registered study. The only thing that was not well described was statistical analysis. Specifically, authors mentioned the use of SEM, HLM, and growth modeling, but it was not clear how these methods would be used to analyze the data. For example, there were four waves of data, but what kinds of changes would be examined? For another example, the authors would look at the relationship between psychosocial demands and antidepressant and opioid use, what types of analysis would be used, correlation or something else? So, this kind of specificity should be clearly described. We know what the authors would examine, but how in terms of specific statistical techniques? A third example, authors stated “Bayesian analysis will be applied to estimate random intercepts and slopes for individuals to understand future time points for which the individual may be at risk for suicidal ideation.” Bayesian analysis is a broad term, and we do not know any specific methods, such as regression or something else in addressing this issue.
---

VERSION 1 – AUTHOR RESPONSE

Reviewer: 1

Dr. Thomas Gerding, University of Cincinnati

Comments to the Author:

This sounds like a very interesting study topic. I am curious whether you can find a relationship between occupational stress exposures and antidepressant / opioid medication use. The reviewed protocol outlines your goals and objectives explicitly and to a degree appropriate for publication.

Reviewer: 2

Prof. Agata Chudzicka-Czupala, SWPS University of Social Sciences and Humanities

Comments to the Author:

Thank you very much for the opportunity to learn about this inspiring, very interesting research project, which was possible through this review. Like the Authors, I am convinced that this study protocol may and should be published in the journal BMJOpen.

I have essentially no substantive or ethical comments on this ambitious research project. I believe that the results of this study can lead to very important conclusions.

Both its objectives, methods, sample and selection of subjects, and the potential benefits of the results are described very clearly here. The literature cited is current and relevant. The Authors have also addressed the ethical issues of the research.

I wish them success in their research.

Reviewer: 3

Dr. Ricard Navinés , IDIBAPS

Comments to the Author:

This is an ambitious study, which using national databases (Australian Workplace Barometer, AWB, Pharmaceutical Benefits Scheme, PBS), wants to answer through a creative method, the relationship between occupational factors, mental health and the use of antidepressants and opioids

Response (CC): Dear reviewer 3, we appreciate your constructive feedback.

Introduction:

In my opinion, the introduction is a bit disorganized. At the end of the second paragraph, an objective of the study is already introduced. It is preferable that all the objectives are discussed together at the end of the introduction, because otherwise it cuts the discourse a bit.

The aim discussed at the end of the second paragraph has been repositioned to adhere to the traditional structure of an introduction.

It begins with the increase in use antidepressants, opioids, corporate climate, then depression, again theories of stress at work, corporate climate... In my opinion this part of the introduction has to be rewritten to make it better understood.

Response (CC): The introduction has been modified for clarity and flow.

Research questions:

In my opinion, the main research question is not fully defined. In fact, the first research question is four. In my opinion, the main research question should be more precisely defined, and perhaps delimited a little more objectives in general, so that it is not confusing or excessive.

Response (CC):

To ensure precision with the main research question we have delimited question as follows;

'Does Psychosocial Safety Climate have a primary role, in process pathways linking working conditions to mental health problems and antidepressant/opioid medication use?'

Confirmatory analysis Plan:

The statistical method to be carried out is not clearly explained

Response (CC): The data plan and statistical method section has been modified to include a clearer depiction of the model development and statistical analyses to be conducted.

Minor suggestions: Apart from the contribution of each author, it is not necessary the authors be named within the protocol. Check the tenses, and write it in the future if the work has not yet been done.

Response (CC): Noted and amended.

Reviewer: 4

Dr. Cody S. Ding, University of Missouri

Dear reviewer 4, we appreciate your constructive feedback.

Comments to the Author:

Overall, this is a well-written manuscript of pre-registered research. The research questions were clear, and the arguments for the study were well articulated. The description of data sources was clear. I do not have major concerns about this pre-registered study.

The only thing that was not well described was statistical analysis. Specifically, authors mentioned the use of SEM, HLM, and growth modeling, but it was not clear how these methods would be used to analyze the data. For example, there were four waves of data, but what kinds of changes would be examined? For another example, the authors would look at the relationship between psychosocial demands and antidepressant and opioid use, what types of analysis would be used, correlation or something else? So, this kind of specificity should be clearly described. We know what the authors would examine, but how in terms of specific statistical techniques?

A third example, authors stated "Bayesian analysis will be applied to estimate random intercepts and slopes for individuals to understand future time points for which the individual may be at risk for suicidal ideation." Bayesian analysis is a broad term, and we do not know any specific methods, such as regression or something else in addressing this issue.

Response (CC): The data plan and statistical method section has been modified to include a clearer depiction of the model development and statistical analyses to be conducted.

Reviewer: 1

Competing interests of Reviewer: No competing interests.

Reviewer: 2

Competing interests of Reviewer: No competing interests

Reviewer: 3

Competing interests of Reviewer: I understand the above, and consent to the named publication of my review.

I confirm I have no conflicts of interest

Reviewer: 4

Competing interests of Reviewer: No competing interests

VERSION 2 – REVIEW

REVIEWER	Navinés , Ricard IDIBAPS
REVIEW RETURNED	16-Nov-2023

GENERAL COMMENTS	I consider the review made by the authors following the reviewers' suggestions to be adequate. I have no further comments to make. The research protocol is much better understood, which I believe will undoubtedly provide valuable information
---

REVIEWER	Ding, Cody S. University of Missouri
REVIEW RETURNED	22-Nov-2023

GENERAL COMMENTS	I am happy with the revision, and the authors explicitly described the purposes of the statistical methods used in the study, which was my only concern before the revision.
--